# Domain Generalization for Retinal Vessel Segmentation with Vector Field Transformer

**Dewei Hu**[1]                                             DEWEI.HU@VANDERBILT.EDU
**Hao Li**[1]                                               HAO.LI.1@VANDERBILT.EDU
**Han Liu**[2]                                              HAN.LIU@VANDERBILT.EDU
**Ipek Oguz**[1,2]                                          IPEK.OGUZ@VANDERBILT.EDU
[1]*Electrical and Computer Engineering and* [2]*Computer Science, Vanderbilt University, Nashville, TN*

**Editors:** Under Review for MIDL 2022

## Abstract

Domain generalization has great impact on medical image analysis as data distribution inconsistencies are prevalent in most of the medical data modalities due to the image acquisition techniques. In this study, we investigate a novel pipeline that generalizes the retinal vessel segmentation across color fundus photography and OCT angiography images. We hypothesize that the scaled minor eigenvector of the Hessian matrix can sufficiently represent the vessel by vector flow. This vector field can be regarded as a common domain for different modalities as it is very similar even for data that follows vastly different intensity distributions. Next, we leverage the uncertainty in the latent space of the auto-encoder to synthesize enhanced vessel maps to augment the training data. Finally, we propose a transformer network to extract features from the vector field. We show the performance of our model in cross-modality experiments. Our code and trained model are publicly available at `https://github.com/MedICL-VU/Vector-Field-Transformer`.

**Keywords:** domain generalization, vessel segmentation, vector field, transformer, data augmentation

## 1. Introduction

Deep learning has become the prevailing solution for many medical image analysis tasks (Li et al., 2021; Hu et al., 2020; Li et al., 2020a) given its remarkable performance. However, as a data-driven algorithm, it is sensitive to the data distribution, especially when the available annotated training set is limited. Unfortunately, most medical image modalities present strong distribution shifts between datasets, caused by the use of various imaging protocols and/or scanner vendors. To tackle this, the ideas of domain adaptation (DA) (Guan and Liu, 2021) and domain generalization (DG) (Wang et al., 2021; Zhou et al., 2021) have been proposed. Suppose a model is trained on data in one or several different but related domains. DA aims to optimize the performance of it on a given target domain. DG is more challenging as the target domain is completely inaccessible during training.

There are three main classes of approaches to increase the generalization capability of a deep model. First, there are data augmentation/generation based methods (Khirodkar et al., 2019; Zhou et al., 2020). By applying hand-crafted perturbations to training data or by leveraging adversarial models to generate new data that is out of the current domain distribution, the training domain is expanded in this approach. Next, there are the representation disentanglement approaches (Xu et al., 2014; Ouyang et al., 2021). Given input

data from various domains, these let the model extract domain-invariant latent features that are transferable across data distributions. This is equivalent to mapping the data following different distributions to a common space. The last family of approaches is using general learning strategies (e.g., meta learning (Li et al., 2018; Balaji et al., 2018)). In this study, we investigate a workflow that marries the first two approaches to achieve DG.

There are many implementations of DG for MRI (Liu et al., 2020; Li et al., 2020a), CT (Khandelwal and Yushkevich, 2020) and fundus photography (Yang et al., 2021; Liu et al., 2021), but the OCT and OCT angiography (OCT-A) are rarely discussed in this context. OCT-A is an important tool to visualize retinal vessels. However, the complex retinal vasculature composed of thin plexus requires huge effort to obtain 3D manual annotation for supervised training. Instead, **our overall goal is to use annotated 2D fundus images to train a network that is capable of vessel segmentation on OCT-A data**. Inspired by our previous work (Hu et al., 2021), we begin by applying an auto-encoder to generate enhanced vessel maps in the latent space. As there is no direct supervision, these enhanced vessel maps are in arbitrarily different styles/contrasts for each re-run of the training process. We use these differing vessel maps as augmentation for our training set. Next, since the human perception of 'vesselness' depends heavily on the local contrast and general shape instead of absolute intensity value, we hypothesize that the minor eigenvector of the Hessian matrix can sufficiently represent the vessels. By introducing the vectors, our goal is to let the model learn the shape of the vessels independently from the image intensity distribution. This would allow us to achieve generalization with regard to the 'style' of the input image. Thus, we convert the image into a vector field which we deem as a common feature space. Then a vector field transformer (VFT) is proposed to extract shape features. Since the transformer leverages the attention mechanism based upon the dot product of feature vectors, it is suitable to work on a vector field. In addition, we designed a parallel pathway with three types of patch sizes that allow the model to see different ranges of context. Our main contributions are:

- Introducing the vector field to generalize different modalities for vessel segmentation.

- Auto-encoder to generate different styles of enhanced vessel maps for augmentation.

- VFT with parallel transformer layers to separate the input with various window sizes.

## 2. Methodology

Our approach is based on the key observation that, despite very different image appearances, the structural/geometrical features of vessels in color fundus photography and depth projected OCT angiography perceptually present strong similarities. However, the existing learning-based approaches for vessel segmentation rarely attempt to bridge the gap between these two modalities. In other words, instead of focusing on the structural patterns of the vessels, the current models are more dependent on intensity distribution.

We model the domain generalization as follows: We denote the number of domains by $\Omega$. Then the source domains that are used for training can be represented as $\mathcal{S} = \{S^i | i = 1, \ldots, \Omega_S\}$, while the target set of domains for testing are $\mathcal{T} = \{T^i | i = 1, \ldots, \Omega_T\}$. Each

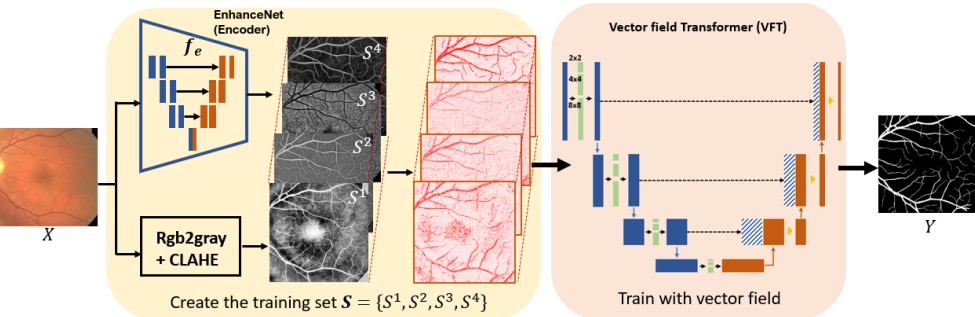

Figure 1: The overall pipeline. The yellow box is the data augmentation used to create the training set $\mathcal{S}$. The pink box shows the segmentation network. In testing, only the pink box is used.

training domain contains $m^i$ pairs of annotated samples $S^i = \{(X^i_j, Y^i_j) | j = 1, \ldots, m^i\}$, where $Y^i_j$ is the label for sample $X^i_j$. Considering the easier availability of manual annotation, we train the model on a fundus dataset (i.e., $\mathcal{S} = \{S^1_{fundus}\}$), then test on two OCT-A datasets (i.e., $\mathcal{T} = \{T^1_{octa}, T^2_{octa}\}$) to demonstrate two different cross-modality scenarios.

To increase the diversity of the training data, we propose a vessel enhancement network (EnhanceNet, Sec. 2.1) to generate enhanced vessel maps that follow a variety of distributions such that the training set $\mathcal{S}$ gets augmented to $\{S^1, \ldots, S^{k+1}\}$, where $k$ is the number of EnhanceNet models trained. This also converts the fundus images into grayscale in the process. Then, to learn the tubular shape of the vessel, we use an intensity-scaled vector field (Sec. 2.2) as the input and propose a multi-size-window transformer to capture the correlation between vectors in a local area. Fig. 1 illustrates this overall DG pipeline.

## 2.1. Data augmentation with vessel enhancement network

The *en-face* projection of OCT angiography is a 2D grayscale image that contains only the vessels. In contrast, the fundus data are color images with relatively poor contrast and also includes other anatomical structures such as the optic disk and the fovea. Some approaches

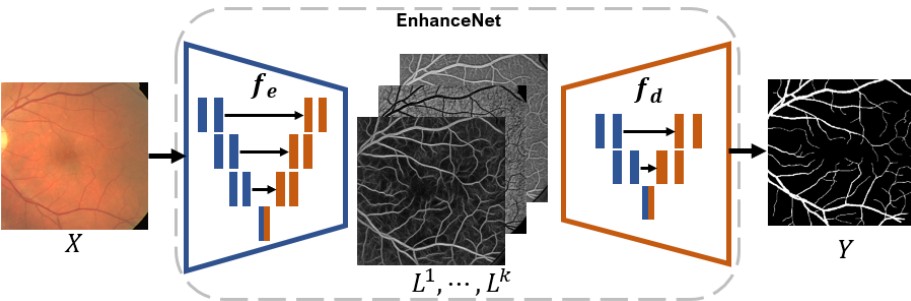

Figure 2: Vessel enhancement network. $X$ is the input color image, $Y$ is the corresponding binary label. $L_i$ denotes the latent space for k different trained models.

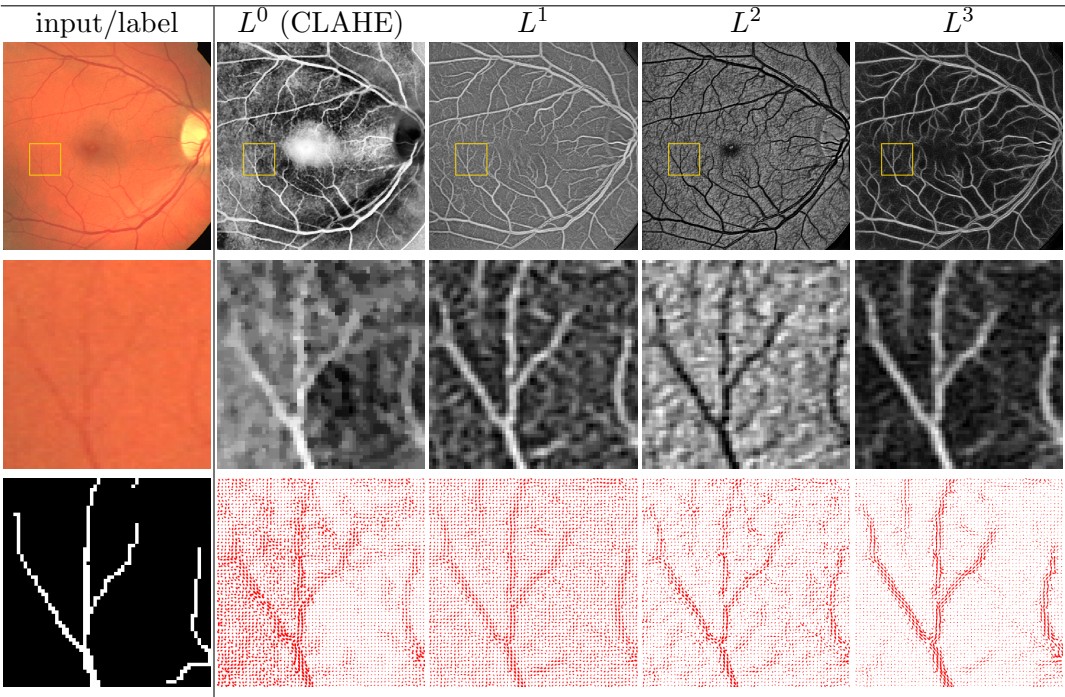

Figure 3: Comparison of EnhanceNet latent images with CLAHE contrast enhancement. The middle row is a zoomed-in view within the yellow box. The bottom row shows the associated vector fields.

convert fundus images to grayscale and apply the contrast-limited adaptive histogram equalization (CLAHE) (Reza, 2004) algorithm to improve the contrast of vessels. However, it is possible for CLAHE to over-enhance noise and/or other structures like macula, such that some vessels are blocked (e.g., the bright middle region in the CLAHE result in Fig. 3). Instead, following the same idea as (Hu et al., 2021), we introduce a vessel enhancement network, EnhanceNet, for data augmentation. It converts the color image to grayscale and simultaneously filters out the irrelevant structures such that the vessels stand out.

The EnhanceNet has an encoder-decoder structure (Fig. 2). Let $X$ be the input color fundus image, and let $Y$ be the corresponding vessel labels. If we view these images as feature sets, $X$ is a larger set that contains both vessel features as well as spurious features (e.g., fovea, optic disk) that are undesirable for the vessel segmentation task. The encoder $f_e$ of EnhanceNet serves as a feature selector that filters these out. If the network output $\hat{Y}$ is a good reconstruction of $Y$, then the latent image $L$ should approximate the intersection of the two sets (i.e., $L \approx X \cap Y$). In order to keep the latent space to the same dimensions as the input image, the residual U-Net architecture is implemented for both the encoder $f_e$ and the decoder $f_d$. Since the target output is the vessel-enhanced latent image, we distribute more parameters in $f_e$ to allow the encoder more flexibility. Given N pixels, the loss function for training is a combination of cross-entropy loss and Dice loss:

$$L_{EnhanceNet} = -\frac{1}{N}\sum_{n=1}^{N} y_n \log \hat{y}_n + \left(1 - \frac{2\sum_{n=1}^{N} y_n \hat{y}_n}{\sum_{n=1}^{N} \hat{y}_n^2 + y_n^2}\right), \qquad (1)$$

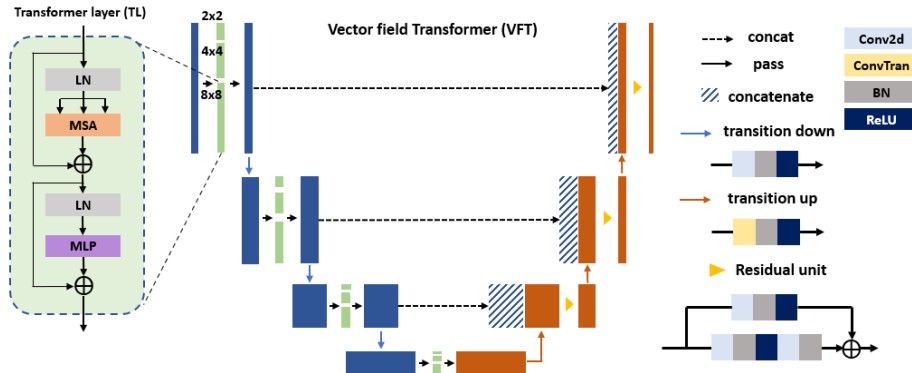

Figure 4: The architecture of the VFT network. Network details are available in Appendix A.

where $\hat{y}_n$ and $y_n$ are the prediction and the ground truth at pixel $n$, respectively.

Since there is no direct supervision over the latent space, the training process can be fairly unstable and the appearance of $L$ is not consistent when the model is re-trained. We take advantage of this additional degree of freedom and set $k = 3$ in our experiment to train 3 different models. This results in 3 different latent images $L^1$, $L^2$ and $L^3$ for each input image $X$, as illustrated in Fig. 3. We note that sometimes the vessel intensity can be flipped between different $L^i$. We also observe that the background texture features are not completely removed in $L^1$ and $L^2$. $L^3$ is closer to the ideal case since background texture is strongly suppressed and only vessels are extracted. It is clear from Fig. 3 that all 3 outputs from EnhanceNet are dramatically cleaner than CLAHE results, and hence better able to simplify the downstream segmentation task. By leveraging the uncertainty of EnhanceNet, we can generate vessel maps in several synthetic domains given a single input image as an effective way to augment the training data. All four types of generated domains are applied in the training stage. If we denote the CLAHE image as $L^0$, then the domains are $S^i = \{(L_j^i, Y_j)\}_{j=1}^m$ where $m = 20$ and $i = 0, 1, 2, 3$ in our case, and the whole training set is defined as $\mathcal{S} = \{S^0, S^1, S^2, S^3\}$.

### 2.2. Vector field transformer

Frangi *et al.* classically model tubular structures with the vesselness measure, defined by the eigenvalues of the Hessian matrix $\mathcal{H}$ (Frangi et al., 1998). Following the same intuition, we observe that the minor eigenvectors of $\mathcal{H}$ form a smooth vector field with streamlines that follow along the retinal vessels. To create the vector field given a grayscale image $X$, we compute the Hessian $\mathcal{H}$ at $X(i, j)$ by 2D convolution with the derivative of a Gaussian with $\sigma = 0.1$. After eigen-decomposition, we keep the second eigenvector $\boldsymbol{v}_2(i, j)$ that corresponds to the smaller eigenvalue, which aligns with the vessel direction. The image intensity is used as the magnitude of the vector, which yields the vector field $V(i, j) = X(i, j)\boldsymbol{v}_2(i, j)$. In our experiments, the image intensities of $X$ are normalized to the $[0, 255]$ range. Note that for cases like $L^2$, we flip the intensity before normalization.

One promising property of this vector field representation is that it allows a consistent appearance for vessels across different image modalities, as it emphasizes struc-

tural/geometrical information rather than pure image intensity. In the last row of Fig. 3, we observe that the vessel is represented by the vector flow formed by vectors sharing coherent orientations and magnitudes. This property holds across diverse domains $L^i$. Segmenting vessels is then equivalent to find these locally clustered vectors that are coherent in magnitude and direction. This is very suitable for the self-attention mechanism to work. By taking the dot product with other vectors, the homogeneity of vector orientation and intensity levels can easily be captured. Hence, we propose a vector field transformer (Fig. 4) that takes the vector field as input and leverages 3 parallel transformer layers (TL) with different window sizes ($2 \times 2$, $4 \times 4$ and $8 \times 8$) to extract features in multiple scales. Since the vector is already regarded as a feature, no further embedding is needed. Similar to TransUNet (Chen et al., 2021), the transformer blocks are only applied in the encoder layers, while the decoder contains residual blocks. The loss function for VFT is cross-entropy.

### 2.3. Datasets

We use three publicly available datasets. **DRIVE.** The DRIVE dataset (Staal et al., 2004) consists of 20 labelled fundus images of size $565 \times 584$. We use these as our training set. **ROSE.** The ROSE dataset (Ma et al., 2020) includes two type of annotations: the centerline-level (sparse) labels of thin vessels, and pixel-level (dense) labels of thick vessels. We use 30 images of size $304 \times 304$ with pixel-level dense labels for testing. **OCTA500.** OCTA500 (Li et al., 2020b) contains two subsets: OCTA_6M and OCTA_3M. We use OCTA_6M, which includes 300 samples with larger field of view (6mm×6mm×2mm). The projection maps of different tissue layers and manual vessel segmentations are available in 2D. We use the OCTA_6M internal limiting membrane (ILM) to outer plexiform layer (OPL) projection as our second testing set (i.e., 300 images with size $400 \times 400$).

### 2.4. Baseline models and implementation details

To evaluate the proposed method, we performed a comprehensive ablation study. First, we train a residual UNet (ResUNet) with the same number of layers as VFT to assess the advantage of using the parallel transformers. Note that both models take the vector field as input. Next, using the same ResUNet structure, we train a model that takes the intensity image as the input to assess whether the vector field representation of the image helps with the recognition of vessel shape. Finally, we train all three models with and without data augmentation (Sec. 2.1).

All networks are trained and tested on an NVIDIA RTX 2080TI 11GB GPU. We use a batch size of 3 and train for 300 epochs. We use the Adam optimizer with the initial learning rate of $1 \times 10^{-5}$ for VFT, $1 \times 10^{-4}$ for Residual UNet. The learning rate for both networks decay by 0.5 every 3 epochs.

### 3. Results

Qualitative results are shown in Fig. 5, where the top two rows are from ROSE and the bottom two rows are from OCTA500. Red and green represent false negatives (FN) and false positives (FP), respectively. Each column compares the model trained with/without augmentation (Sec. 2.1). The effects of the vector field input and the transformer are

| Algorithm | DSC | | ACC | | SEN | | SPE | |
|---|---|---|---|---|---|---|---|---|
| | w/o | w | w/o | w | w/o | w | w/o | w |
| ResUNet(int) | 0.6705 ±0.0239 | **0.6912** **±0.0364** | 0.8872 ±0.0092 | 0.9129 ±0.0128 | **0.7542** **±0.0681** | **0.6419** **±0.0898** | 0.9130 ±0.0141 | 0.9641 ±0.0102 |
| ResUNet(vec) | 0.7031 ±0.0261 | 0.6510 ±0.0299 | 0.9220 ±0.0109 | 0.9134 ±0.0145 | 0.6837 ±0.0713 | 0.5256 ±0.0424 | 0.9550 ±0.0077 | 0.9846 ±0.0035 |
| VFT | **0.7602** **±0.0244** | 0.6807 ±0.0288 | **0.9303** **±0.0110** | **0.9192** **±0.0136** | 0.7221 ±0.0633 | 0.5604 ±0.0446 | **0.9694** **±0.0051** | **0.9851** **±0.0032** |

Table 1: ROSE results. DSC: Dice score, ACC: accuracy, SEN: sensitivity. SPE: specificity. Bold: best score per column. w, w/o: with or without augmentation.

| Algorithm | DSC | | ACC | | SEN | | SPE | |
|---|---|---|---|---|---|---|---|---|
| | w/o | w | w/o | w | w/o | w | w/o | w |
| ResUNet(int) | 0.6344 ±0.0532 | 0.7165 ±0.0373 | 0.9069 ±0.0124 | 0.9494 ±0.0070 | 0.8852 ±0.0422 | 0.7055 ±0.0795 | 0.9098 ±0.0156 | 0.9751 ±0.0104 |
| ResUNet(vec) | 0.7005 ±0.0398 | 0.7754 ±0.0257 | 0.9343 ±0.0074 | 0.9604 ±0.0064 | 0.8455 ±0.0508 | 0.7460 ±0.0373 | 0.9440 ±0.0097 | **0.9827** **±0.0048** |
| VFT | **0.7365** **±0.0446** | **0.7876** **±0.0291** | **0.9414** **±0.0069** | **0.9610** **±0.0060** | **0.9045** **±0.0337** | **0.7936** **±0.0458** | **0.9455** **±0.0083** | 0.9785 ±0.0057 |

Table 2: OCTA500 results. DSC: Dice score, ACC: accuracy, SEN: sensitivity. SPE: specificity. Bold: best score per column. w, w/o: with or without augmentation.

shown left-to-right. The model trained on the intensity input gets many FPs, and vessel thicknesses are overestimated. The former issue is largely resolved by using the vector field as input. By training on augmented data, the model becomes less aggressive and thin vessels are no longer over-dilated. For the ROSE dataset, the labeled vessels are relatively thick which are easy to capture, so even the vanilla ResUNet works well. We note that, although the accuracy of the segmentation raises with augmentation, this may induce a drop in sensitivity (Tab. 1) in this dataset. A hypothesis for why ROSE does not benefit from data augmentation is that augmented images have relatively high vessel intensity, while the small vessels in ROSE appear darker. For the OCTA500 dataset, in which the thinner vessels are also labeled, the proposed method gets the best outcome with regard to almost all metrics (Tab. 2). For both datasets, VFT performs better with vector input.

## 4. Conclusion

The domain generalizability of a deep learning model is essential in medical image analysis. In this work, we explore mapping vessel images from different modalities to a common space by creating the eigenvector field from Hessian matrices. Then, we set up a vector field transformer to capture the structural features modelled by the correlation between vectors.

| Input/label | ResUNet (intensity) | ResUNet (vector) | VFT |
| --- | --- | --- | --- |

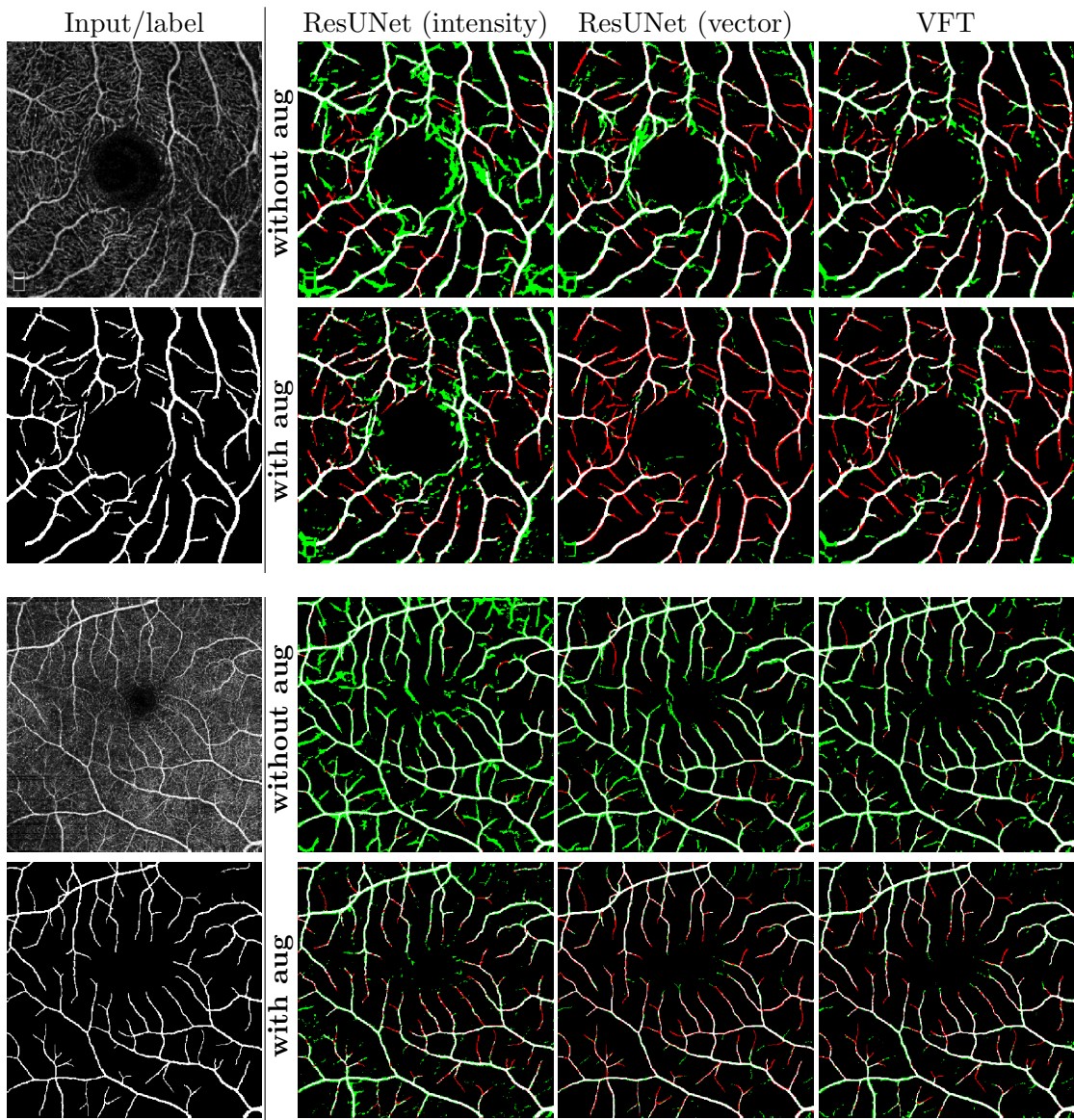

Figure 5: Effects of input (intensity/vector), model (ResUNet/VFT) and data augmentation. **Red** and **green** represent false negatives and false positives, respectively. The top two rows are from ROSE, the bottom two rows are from OCTA500.

This makes promising gains in Dice score. Moreover, we leverage the uncertainty of the latent output from EnhanceNet to augment the data with 3 different synthetic domains. This further improves the segmentation accuracy. Our approach can be extended to other segmentation tasks in tubular objects (e.g., airway trees) in future research.

## Acknowledgments

This work is supported by Vanderbilt University Discovery Grant Program.

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

## Appendix A. VFT architecture

**Residual U-Net.** The backbone of the neural network is a residual U-Net that takes the input vector field with shape $x^1 \in \mathbb{R}^{2 \times 256 \times 256}$. Herein, the superscript denotes the input of the first layer. Note that the residual units are replaced by paralleled transformer layers in the compression pipeline. These transformer layers will not change the dimension of the input tensor. Details are provided in the following paragraph. The downsampling is achieved by a transition down-block that contains a 2d convolution layer, a batch normalization layer, and an exponential linear unit (ELU). Intuitively, this transition down-block will increase the channel number while reducing the height and width of the image by half. In our experiment, we apply a 5-layer model with channel number $\{2, 8, 16, 32, 64\}$ (i.e., $x^5 \in \mathbb{R}^{64 \times 16 \times 16}$). In the decoder part, we apply the transpose 2d convolution layer, batch normalization, and ELU in the transition up-block.

**Transformer layers.** We incorporate the transformer blocks in each layer of the residual U-Net encoder. In order to capture the vector orientation similarity in different scales of context, we break the image into three types of patches ($2 \times 2$, $4 \times 4$, $8 \times 8$). This forms the three paralleled transformer layers (TL). If the input $x \in \mathbb{R}^{C \times H \times W}$, then the output of each TL has the same dimension (i.e., $TL(x) \in \mathbb{R}^{C \times H \times W}$). We concatenate all three outputs by channel and apply a 2D convolution to linearly map it back to $\mathbb{R}^{C \times H \times W}$. We leverage the same transformer layer structure as proposed in (Chen et al., 2021). We set the number of heads in the multi-head self-attention layer (MSA) to $\frac{C}{2}$. The output of the multi-layer perceptron is set to be $4C$.

Unlike other implementations of transformers in medical image analysis (Hatamizadeh et al., 2022), VFT focuses on the context within the partitioned patches instead of the potential correlation between windows. Therefore, no feature embedding is required for VFT as the eigenvectors already represent the structural pattern. The transformer layer acts on the vectors within the patch to extract the vector similarity. Intuitively, the vectors within a vessel in a small patch should be homogeneous in both orientation and magnitude.

## Appendix B. Comparison with same modality segmentation

To further validate the cross-modality segmentation performance, we compare it with a VFT model trained and tested on OCT-A images. There are 300 subjects in OCTA500 6M dataset. We use 200 of them as training data, while the rest serves as testing data. The following table shows the comparison between the two settings, i.e., same modality vs. cross-modality:

| train/test | DSC | ACC | SEN | SPE |
|---|---|---|---|---|
| OCT-A/OCT-A | 0.8200 ±0.0287 | 0.9686 ±0.0065 | 0.7817 ±0.0430 | 0.9881 ±0.0046 |
| fundus/OCT-A | 0.7876 ±0.0291 | 0.9610 ±0.0060 | 0.7936 ±0.0458 | 0.9785 ±0.0057 |

Table 3: Performance of the VFT model in the same modality vs. cross-modality settings.

The gap between the two Dice scores (0.7878 vs. 0.8200) is modest, and the other metrics are similarly comparable. Hence, this experiment supports the conclusion that the vector field substantially facilitates cross-modality generalization.

