# OpenReview forum: "Domain Generalization for Retinal Vessel Segmentation with Vector Field Transformer"
_MIDL.io/2022/Conference — MIDL 2022_

### Official Review · Reviewer_Lofe · 2022-01-22

**Confidence:** 2
**Preliminary Rating:** 4
**Recommendation:** Oral

**Summary:**

The paper proposed domain adaption method for the task of fundus and OCT image vessel segmentation. Three public benchmarks are used to evaluate the performance such as The DRIVE dataset (Staal et al., 2004), which consists of 20 labelled fundus images
The ROSE dataset (Ma et al., 2020), which includes two type of annotations: the centerline-level (sparse) labels of thin vessels, and pixel-level (dense) labels of thick vessels. And OCTA500 (Li et al., 2020b), which contains two subsets: OCTA 6M and OCTA 3M.

**Strengths:**

1. Novelty. The design of hypothesizing that the minor eigenvector of the Hessian matrix sufficiently represents the vessels looks interesting to me.

2. Sufficient evaluation process with the statistical performance provided.

**Weaknesses:**

I did not see much weakness in this manuscript, except that the number of compared methods in Table 2 are limited. Previous domain adaption methods are encouraged to be presented in the revision to strengthen the whole paper.


**Deanonymize Review:**

no

**Final Rating After The Rebuttal:**

5: Strong Accept

**Justification Of The Final Rating:**

The author addressed all my questions; thus, I raised my score to strongly accept. The author gave a detailed explanation about compared works and promises to add more in the journal version, and the code will be made available.

**Paper Type:**

methodological development

**Questions To Address In The Rebuttal:**

I did not see much weakness in this manuscript, except that the number of compared methods in Table 2 are limited. Previous domain adaption methods are encouraged to be presented in the revision to strengthen the whole paper.


1. Will the code be made publicly available in the future to benefit the whole community's development?



**Special Issue:**

no

---

### Official Review · Reviewer_Rt2n · 2022-01-24

**Confidence:** 5
**Preliminary Rating:** 4
**Recommendation:** Oral, Poster

**Summary:**

The paper addresses the problem of domain generalization in vessel segmentation in color fundus and OCTA images. The authors propose to use the second eigenvector of Hessian matrix as a common feature, and corresponding vector space as a common feature space. The authors introduce the vector field transformer model to segment the vessel using the input images converted to the proposed feature space. To additionally augment the training data, the EnchanceNet is employed.

**Strengths:**

- a novel application of the hessian filter.
The idea of using the Hessian matrix, its eigenvalues, and eigenvectors for vesselness analysis is not novel and was explored in the past, for example [1]. However, the idea of using the Hessian eigenvector as a base for the common feature space between different domains (CF and OCTA) is novel and hasn't been yet explored.
- the open datasets. The authors use public datasets, which increases the reproducibility.



1) N. M. Salem, S. A. Salem and A. K. Nandi, "Segmentation of retinal blood vessels based on analysis of the hessian matrix and Clustering Algorithm," 2007 15th European Signal Processing Conference, 2007, pp. 428-432.

**Weaknesses:**

- missing experiments.
The experiments of a model trained on OCTA and tested on OCTA are missing ( the oracle model). This would mark the best performance the domain generalization methods can achieve. Yet it's not clear how well the proposed method performs compared to the pure OCTA model. If the performance gap is modest, it would add increase the value of the proposed method.

- questionable results on the ROSE dataset. The authors argue that the drop in performance after applying augmentation is caused by the difference in intensities. However, the common normalized vector space is designed to be invariant and move from the intensities. The low-intensity vessels and high-intensity vessels should produce similar vector fields to some extent. Thus the augmentation should cripple the ResUNet(int) due to the intensities issue described by the authors, and improve vector field-based methods. The experiments demonstrate the opposite. It requires in-depth explanation.

- missing implementation details. The details on the VFT are missing. The authors mention "Similar to TransUNet the transformer blocks are only applied in ...". Is the architecture identical to TransUNet? Or only this mentioned aspect. Section 2.2 requires revision and extension.

**Deanonymize Review:**

no

**Detailed Comments:**

The last paragraph of the introduction is confusing. Moving from the high-level details the authors start describing the implementation of augmentation. 2 out of 3 contributions are compressed into two lines of the last paragraph when the details on data augmentation occupy 7 lines. This paragraph should be reworked.

**Final Rating After The Rebuttal:**

4: Weak Accept

**Justification Of The Final Rating:**

I thank the authors for the detailed response. The authors addressed my concerns in the revision and extended their validation by adding additional experiments to the supplementary materials. I believe this paper is an accept.

**Paper Type:**

methodological development

**Questions To Address In The Rebuttal:**

Questionable results on the ROSE dataset described in the Weaknesses section required discussion.

The VFT description should also be improved since it is one of the main contributions and it takes 1/2 page.


**Special Issue:**

yes

---

### Official Review · Reviewer_crYe · 2022-01-24

**Confidence:** 4
**Preliminary Rating:** 5
**Recommendation:** Oral

**Summary:**

This article proposes strategies to train a 2D retinal vessel segmentation network that tends to be domain-independant.
The strategy rely on two steps: first the training set is augmented by applying an autoencoder that generates several enhanced vessel maps from one input image; then, the input images are transformed to vector fields, that are supposed to be contrast-independent, and fed to a vector field transformer (VTF) network designed to perform the segmentation task  based on transformers layers.
The authors validated this approach based on 3 datasets and an ablative study showing the interest of the vector field input and of the VTF architecture.


**Strengths:**

The article is well written and tackles a very interesting problem for the community in vessel segmentation.

The vessel enhancement network is a very clever and simple idea that may be useful for domain generalization in many applications.

The input transformation to vector field using the smaller eigenvalue of the Hessian is also an interesting idea for domain generalization.

The ablative study to evaluate the interest of the vector field input and VTF architecture.

**Weaknesses:**

The VFT architecture description lacks details. In its current form, it is not very clear, which prevents the reader from fully understanding the approach (see detailed comments). Moreover, the authors do not provide their code, which also reduce the reproducibility of their work.

The interest of both the VFT and vessel enhancement network at the same time is not very clear on the ROSE dataset.

The authors should have tried to train their approach on another dataset than drive and test it on the two others.

**Deanonymize Review:**

no

**Detailed Comments:**

The VTF architecture should be detailed in order to be reproducible. In particular, the authors should :
- Detail the dimension of each layer in Figure 4.
- Define each block in the transformer layer (TL).
- Explain briefly the principle of the TL.
- Explain what is performed after applyin the 3 TL with different scale (concatenation ?).

The ablative study could include an experiment on the interest of the multi-scale  TL.

Two sentences are not clear and should be rephrased :
- page 3: "We denote the \omega as the number of ..."
- page 4 : "Let X be the input color fundus image, ..."



**Final Rating After The Rebuttal:**

5: Strong Accept

**Justification Of The Final Rating:**

I thank the authors for answering my questions. The description of the VFT + the future GitHub repo makes the work more reproducible and will surely be useful to the community. I still believe that it is a promising work and I look forward to the journal version with more experiments and validations on other datasets.

**Paper Type:**

methodological development

**Questions To Address In The Rebuttal:**

The authors should clarify the VTF architecture (see above).

Why the authors did not train their approach on another dataset than drive and test it on the other two ? This is in my opinion a very important experiment to validate that the approach is able to generalize from any source domain.

The authors should provide their code.


**Special Issue:**

yes

---

### Official Review · Reviewer_pECV · 2022-01-27

**Confidence:** 4
**Preliminary Rating:** 2
**Recommendation:** Poster

**Summary:**

This paper proposes a domain generalization framwork for retinal vessel segmentation. The overall goal is to use annotated 2D fundus photography to train a network that is capable of vessel segmentation on OCT-A angiography images. Different from existing models dependent on intensity distribution, the proposed approach focuses on the structural patterns of the vessels, introducing the vector field to generalize different modalities for vessel segmentation.


**Strengths:**

1. Authors Introduce the vector field to generalize different modalities for vessel segmentation.
2. Authors use an auto-encoder to generate different styles of enhanced vessel maps for data augmentation as multi-source domain .


**Weaknesses:**

 1. This paper only performs a comprehensive ablation study to assess the advantage of using the parallel transformers and whether the vector field representation of the image helps with the recognition of vessel shape, respectively. This paper lacks a comparison with other methods of domain generalization.
2. Authors claim proposed method is only suitable for segmentation tasks in tubular objects.


**Deanonymize Review:**

no

**Detailed Comments:**

It is recommended to add comparative experiments with other domain generalization methods to prove the superiority of the method under the same setting.

**Final Rating After The Rebuttal:**

4: Weak Accept

**Justification Of The Final Rating:**

This paper proposes a domain generalization framework for retinal vessel segmentation. The overall goal is to use annotated 2D fundus photography to train a network that is capable of vessel segmentation on OCT-A angiography images. Different from existing models dependent on intensity distribution, the proposed approach focuses on the structural patterns of the vessels, introducing the vector field to generalize different modalities for vessel segmentation. However, this paper still lacks comprehensive comparative experiments with other methods of domain generalization.

**Paper Type:**

both

**Questions To Address In The Rebuttal:**

1. Why this paper does not compare with other existing methods of domain generalization?
2.  Is the proposed method only suitable for segmentation tasks in tubular objects? Please discuss the limitations of this paper.


**Special Issue:**

no

---

### Meta-Review · Area_Chair_AgxH · 2022-02-19

**Recommendation:** Accept (Oral)
**Confidence:** 5

**Metareview:**

The authors have clearly clarified the issues raised by the reviewers. Overall, all reviewers are satisfied with the response given by the authors and are glad to see that the quality of the paper has been improved substantially. The rebuttal looks reasonable and correct to me.

---

### Decision · Program_Chairs · 2022-02-28

Accept